# ORTHOGONAL SEQUENTIAL FUSION IN MULTIMODAL LEARNING

## ABSTRACT

The integration of data from multiple modalities is a fundamental challenge in machine learning, encompassing applications from image captioning to text-to-image generation. Traditional fusion methods typically combine all inputs concurrently, which can lead to an uneven representation of the modalities and restricted control over their integration. In this paper, we introduce a new fusion paradigm called Orthogonal Sequential Fusion (OSF), which sequentially merges inputs and permits selective weighting of modalities. This stepwise process also enables the promotion of orthogonal representations, thereby extracting complementary information for each additional modality. We demonstrate the effectiveness of our approach across various applications, and show that Orthogonal Sequential Fusion outperforms existing fusion techniques in terms of accuracy, while also providing valuable insights into the relationships between all modalities through its sequential mechanism. Our approach represents a promising alternative to established fusion techniques and offers a sophisticated way of combining modalities for a wide range of applications, including integration into any complex multimodal model that relies on information fusion.

## 1  INTRODUCTION

Data integration across various modalities is a foundational pillar in machine learning and artificial intelligence. Central to this concept is the ambition to augment our understanding of complex and heterogeneous data, broadening its applications from straightforward unimodal tasks to more sophisticated endeavors such as the encoding and processing of comprehensive surrounding data in the context of human-computer interactions (Jaimes & Sebe, 2007) or autonomous driving (Xiao et al., 2020). By concurrently processing and understanding different types of data, multimodal learning promises to yield more robust models that provide a comprehensive interpretation of data and result in more accurate predictions and insights (Ngiam et al., 2011).

The ubiquity of big data in multimodal machine learning, be it text, image, audio, video, or other types, has underscored the importance of effective fusion techniques (Baltrušaitis et al., 2018). However, realizing this potential is not without its challenges. Current techniques, while numerous and varied, often struggle to adequately represent different modalities in an optimized manner. Most fusion methods process all the inputs in a single step, which leads to a rigid integration that can inadvertently emphasize or ignore certain modalities. This concurrent processing could result in the dominance of one modality over the others, leading to an uneven representation and a limitation in capturing the full diversity of the information that the different inputs offer.

In response to these challenges, our work introduces Orthogonal Sequential Fusion (OSF), an innovative fusion paradigm aimed at overcoming the limitations of existing multimodal fusion techniques. Unlike traditional fusion methods, OSF operates in a sequential manner, integrating one modality at a time, thereby offering a flexible way to weight and prioritize individual modalities based on their relevance. This approach ensures a balanced representation and maximizes the extraction of complementary information across modalities via its custom loss function. OSF's stepwise process promotes orthogonal representations, effectively capturing the distinct and complementary information that each modality offers. This not only enhances the overall performance of the system but also provides valuable insights into the relationships and interactions between different modalities, a property, to our knowledge, unprecedented in existing model-agnostic fusion techniques.

In this paper, we carry out an extensive series of experiments to illustrate the effectiveness of OSF in addressing a multitude of data integration scenarios. These experiments are carefully designed to encompass a broad spectrum of applications, including variations in the number of modalities and variation in model complexity. Moreover, we provide evidence of the practical effectiveness of OSF through its integration within a state-of-the-art model, highlighting its ability to enhance performance beyond traditional fusion techniques in a specific application. These experiments benchmark OSF against established fusion techniques, clearly illustrating that our approach outperforms these methods in terms of evaluation metric.

The contributions of this paper can be summarized as follow:

- We introduce OSF, a novel fusion paradigm for multimodal machine learning. This approach allows for the stepwise integration and discretionary ordering of modalities, promoting a balanced representation and optimizing the extraction of complementary information from each modality.
- We benchmark OSF against prevailing fusion approaches, on diverse datasets, and demonstrate that OSF consistently surpasses the performance of most model-agnostic fusion techniques, exhibiting a particularly notable superiority within the context of a highly multimodal environment.
- We incorporate OSF into a state-of-the-art model, illustrating its seamless integration even within highly sophisticated and complex models. In addition, our findings highlight the ability of this fusion paradigm to further boost performance in already high-achieving applications.

## 2 RELATED WORK

Within the landscape of machine learning research, the integration of data from diverse modalities is an increasingly significant area. This concept, broadly referred to as multimodal learning, is founded on the principle that leveraging multiple sources of data can reveal a more comprehensive and nuanced understanding of the problem at hand than would be possible from any single modality individually (Ngiam et al., 2011). However, the success of multimodal learning is contingent on the effective fusion of these data streams, a process that presents a unique set of challenges and considerations (Baltrušaitis et al., 2018).

**Multimodal Learning and Traditional Fusion Techniques.** Traditional fusion methods have found widespread application in a variety of fields, spanning from computer vision and speech recognition to natural language processing. These techniques, including early, late, and hybrid fusion methods, represent the foundational bedrock of multimodal fusion and have significantly influenced contemporary research. Early fusion, or feature-level fusion, merges the features of all modalities at the very beginning of the process (Barnum et al., 2020). Conversely, late fusion, often referred to as decision-level fusion, initially processes each modality separately, then combines these individual decisions to form a final conclusion (Ramirez et al., 2011). Hybrid fusion, a balanced blend of early and late fusion, leverages the strengths of both methodologies, offering a more versatile approach to multimodal data integration (Lan et al., 2014). Even in the face of increasingly complex models and methodologies, these traditional fusion techniques remain prevalently employed, often serving as the base upon which these more advanced models are built (Ramachandram & Taylor, 2017).

**Limitations of Traditional Fusion Methods.** While traditional fusion methods still hold a prominent place in multimodal learning, they are not without certain constraints. One of the limitations of these approaches is their proneness to process all modalities concurrently. This simultaneous processing can lead to an over-representation of dominant modalities and an under-representation of subtler ones, which can contain valuable information. These imbalances can weaken the overall efficacy of the fusion process and contradict the core aim of multimodal learning: to assimilate and harness the unique insights provided by complementarities and redundancies of the information conveyed by the modalities. Furthermore, traditional fusion techniques lack a mechanism for dynamic weighting or prioritization of individual modalities, a shortfall that becomes increasingly significant when dealing with data that is either imbalanced, noisy or highly multimodal. These identified limitations emphasize the need for fusion methodologies that offer a more adaptive and refined approach.

**Recent Advancements in Multimodal Fusion.** In recent years, efforts to address the shortcomings of conventional fusion techniques have encouraged the development of novel approaches to reshape the fusion process. A significant breakthrough in this area has been the introduction of attention mechanisms (Yu et al., 2019), enabling the fusion process to dynamically assess the importance of different modalities. This development allows the model to adapt to the unique attributes of each data instance, thereby tailoring its fusion strategy to optimize information extraction. Another prevalent focus revolves around creating shared or joint representations across modalities with the goal of establishing a more unified and integrated fusion process (Wang et al., 2020). Nevertheless, these advancements, albeit significant, have not entirely resolved the task of achieving a balanced and comprehensive representation of several modalities. Moreover, these approaches often lack flexibility, being typically model-based (Baltrušaitis et al., 2018), and may exhibit substantial computational resources and complexity in integration. These methodologies may also fail to fully leverage the complementarities and redundancies among modalities. These enduring challenges underline the ongoing need for further innovative thinking and exploration in the field, towards creating fusion techniques that are both efficient and flexible in dealing with multimodal data.

**Orthogonality in Machine Learning.** A promising direction for multimodal learning lies in the concept of orthogonality, a mathematical principle that has found diverse applications within machine learning research. Encouraging orthogonal representations can minimize redundancy and promote diversity within the feature space (Saxe et al., 2019), qualities that are especially desirable within the context of multimodal fusion. By ensuring that each modality contributes unique and non-redundant information to the fusion process, we can harness the full potential of multimodal data. Orthogonality has been employed to enhance various machine learning tasks, such as initialization (Saxe et al., 2013) and representation learning (Hyvärinen & Oja, 2000). However, its application in multimodal fusion, although very promising (Braman et al., 2021), is largely unexplored.

# 3 ORTHOGONAL SEQUENTIAL FUSION

## 3.1 SEQUENTIAL FUSION

In multimodal fusion, the prevalent practice is to employ all modalities simultaneously for combining the extracted information from disparate sources. However, in this research work, we support the idea of sequentially fusing the modalities, step by step in a pairwise fashion. This sequential approach provides enhanced control and sophistication over the fusion process. Our inspiration for this process is drawn from the human reasoning mechanism, where when presented with three information sources, we can split the sources in groups to arrive at a well-informed decision (Tversky & Kahneman, 1974). Following the same logic, our method fuses the first two modalities together and then merge the resulting information with a third modality. We repeat this process until all the modalities have been processed. A pseudo code for pair-wise sequential fusion is given in Procedure 1. While our research primarily focuses on pair-wise fusion, it's possible to employ sequential fusion for merging more than two modalities at each fusion step.

In addition to offering better control over the modalities' fusion, the proposed sequential algorithm allows for incorporating expert knowledge by postulating which ordering should be beneficial to the model and thus deciding the ordering *a priori*. By selectively combining modalities in a sequential manner, we can effectively exploit the underlying relationships between them and weigh their contribution to the final output accordingly by facilitating the extraction of complementarities and redundancies at each layer. By processing step by step, this approach allows to select which modalities are the most likely to perform well when fused together. This is an important difference compared to the all-at-once fusion methods. This approach enhances the interpretability and transparency of the fusion process and enables us to better understand the underlying data characteristics and avoids imbalanced representation of the modalities. Additionally, sequential fusion can handle varying data scales and types while avoiding the pitfalls of high-dimensional feature spaces.

## 3.2 MODALITIES ORDERING

Determining the optimal order for sequential fusion is a critical step in achieving effective fusion. While the most obvious approach is to test all the possible combinations for ordering the modalities,

---

**Algorithm 1** Sequential Fusion algorithm for pairwise fusion

---

**Input:** $N$ different input modalities $i_1...i_N$ encoded in the same dimension $D$
**Output:** 1 final embedding for all fused modalities of dimension $D$

    $x \leftarrow i_1$
    $y \leftarrow i_2$
    **while** $3 \leq k \leq N + 1$ **do**
        $x \leftarrow \text{FUSE}(x, y)$   ▷ In our case FUSE is CONCAT followed by an embedding in dimension $D$
        $y \leftarrow i_k$
    **end while**

---

it is clear that this method can be computationally expensive and may not scale well as the number of modalities grows. In contrast, our ranking-based approach offers a simple and efficient solution for determining the fusion order while avoiding the computational burden of exhaustive search.

Our approach is based on the idea of training unimodal models and ranking the modalities based on their individual performances. This approach allows us to form an initial ordering of the modalities based on their relative strengths in the task at hand. We then fuse the modalities starting from the least performing modality and moving towards the most performing one. By doing so, we aim to extract complementary information from each modality and fuse them in a way that improves overall performance.

However, there are limitations to this approach. While unimodal training provides an indication of the individual performance of each modality, it may not fully capture the interactions and dependencies that exist between pairs of modalities. As a result, the ranking obtained by this approach may not always lead to the most optimal fusion order, and other methods such as exhaustive search may be necessary to explore all possible fusion orders. Moreover, our approach assumes that the modalities have distinct contributions to the task at hand and can be ranked based on their individual performances. In some cases, certain modalities may be highly correlated or redundant, and the ranking may not accurately reflect their true contributions to the task. In such cases, alternative methods for determining fusion order, such as using expert knowledge, may be more suitable.

Despite these limitations, our ranking-based approach offers a practical and efficient solution for determining fusion order, especially when the number of modalities is high. In our experiments on the CMU-MOSI dataset (Zadeh et al., 2016) with three modalities, we found that the ordering determined by our approach outperformed all other possible orders. More research is needed to investigate the effectiveness of our approach in other domains and with a larger number of modalities.

While our ranking-based approach prioritizes efficiency and scalability, it's worth noting that conducting an exhaustive search over all possible orders of modalities can offer unique insights into the data itself. Specifically, an exhaustive search can reveal synergies between modalities that may not be evident when evaluating them in a unimodal fashion. For instance, two modalities with suboptimal individual performance might, when fused early in the sequence, yield superior results due to complementary features or interactive effects. This kind of nuanced understanding of the relationships between modalities serves as a form of interpretability within the model, helping to clarify not just which modalities are most valuable, but also how they best complement each other to achieve optimal performance.

### 3.3 ORTHOGONAL LOSS FUNCTION

One of the major novelties of our proposed sequential fusion method lies in the loss function. While the sequential fusion process already allows us to combine modalities in groups and extract information at each step, we believe that it could be even more interesting to incentivize the modalities to have complementary information. This is especially true when some modalities have overlap in the information they encode initially. In order to learn useful complementary features through the fusion process, we propose encouraging the latent representation to be orthogonal before fusion occurs.

OSF is inspired by L1-L2 regularization (Zou & Hastie, 2005) and can be applied to any loss function. The objective of OSF is to encourage the fused embeddings at each layer to be orthogonal, which in turn helps to capture more diverse and complementary information from the modalities. To

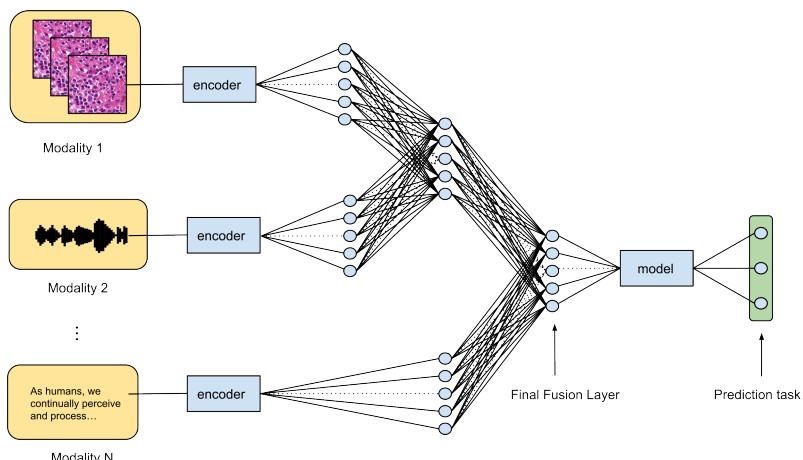

Figure 1: Orthogonal Sequential Fusion with N modalities for a given prediction task

achieve this, we introduce two hyperparameters $\lambda_{1,i}$ and $\lambda_{2,i}$, which control the strength of the L1 and L2 regularization terms, respectively.

Assuming we have N modalities and N-1 fusion layers $L_i = [embd_{1,i}, embd_{2,i}]$, the loss function for OSF can be written as:

$$Loss = \mathcal{L}(preds, outputs, T) + \sum_{i=1}^{N-1} \lambda_{1,i}|embd_{1,i} \cdot embd_{2,i}| + \lambda_{2,i}(embd_{1,i} \cdot embd_{2,i})^2 \quad (1)$$

where $\mathcal{L}$ is the loss function for a given task T, and $preds$, $outputs$ are the predictions from the model and the ground truth, respectively.

The regularization terms in the loss function encourage the embeddings at each layer to be orthogonal. The first term enforces the absolute value of the dot product of the embeddings to be close to zero, while the second term encourages the square of the dot product to be close to zero. Together, these terms ensure that the embeddings are not only diverse but also complementary.

The OSF loss function can be optimized using standard backpropagation algorithms. During training, the network learns to weigh the modalities in a way that maximizes the diversity and complementarity of the fused embeddings at each layer, resulting in a more effective and robust fusion process.

Overall, the use of the OSF loss function in our sequential fusion approach further improves the extraction process for each sequential step. By encouraging orthogonal embeddings, we are able to capture more diverse and complementary information from the modalities, resulting in better performance and more effective models. Our experiments demonstrate the effectiveness of the OSF approach in various applications, as shown in the following section.

## 4 EXPERIMENTS

This section aims to empirically validate the performance of our proposed OSF approach. We conduct experiments on two datasets — CMU MOSI (Zadeh et al., 2016) for multimodal sentiment analysis and TCGA KIRP (Tomczak et al., 2015) for survival prediction — to compare our method against traditional fusion techniques. Additionally, we assess its efficacy when integrated into an existing multimodal model.

## 4.1 DATASETS

We use two multifaceted datasets to thoroughly evaluate the versatility of OSF across various tasks:
**CMU-MOSI** The CMU MOSI dataset is a benchmark in multimodal sentiment analysis, featuring 2199 video segments combining three modalities: audio, visual, and textual data. It is widely employed for both fine-grained (7-class) and coarse-grained (2-class) sentiment classification, as well as regression tasks. The class labels and regression scores are derived from a linear scale that generates scores ranging from -3 to +3.
**TCGA-KIRP** The TCGA KIRP database is commonly utilized for prediction tasks in kidney renal papillary cell carcinoma. From this database we build a dataset with up to 6 modalities, including images, clinical data, mRNA, miRNA, DNAm and copy number variation. We use this dataset to demonstrate the applicability of our fusion approach in a highly multimodal, clinical setting.

## 4.2 METRICS

**2-Class and 7-Class Classification:** We use accuracy as the primary evaluation metric. Accuracy measures the model's capability to classify sentiment. However, it is important to note that the 7-class task inherently involves greater complexity, thus making high accuracy rates more significant.
**Regression:** For the regression task, we adopt Mean Absolute Error (MAE) as our metric. MAE measures the average absolute differences between the predicted and actual values, offering an intuitive understanding of the model's prediction quality in a continuous setting.
**Survival prediction:** The concordance index (c-index) is used to evaluate the effectiveness of our model in survival analysis. It measures the ability of the model to correctly rank pairs of individuals based on their survival times. To further validate our model's reliability in survival prediction, we also calculate the Integrated Brier Score (IBS), which quantifies the prediction error over time.
**Robustness Evaluation:** To assess the robustness of our approach, all experiments are run using multiple random seeds. We report the average and standard deviation of each metric to provide a comprehensive view of the model's performance and volatility across different initializations.

## 4.3 IMPLEMENTATION DETAILS

**Baselines:** For the CMU MOSI dataset, we comply with data processing guidelines established by Liang et al. (2021). The primary objective is to evaluate the effectiveness of our OSF approach. We compare OSF against conventional multimodal fusion methods, including Early fusion, Late fusion, Mean fusion, Max fusion, and Sum fusion as described in Appendix A. Due to the simplicity of the models and the relatively low computational cost, we train separate models for each prediction task on the CMU MOSI dataset. Each modality, represented as a sequence, is initially processed by a Gated Recurrent Unit (GRU) with tanh activation. This is followed by a dense layer with ReLU activation and we use a dropout rate of 0.1. To enable fair comparisons and maintain model simplicity, an embedding dimension of 32 is used for input modalities in both the CMU MOSI and TCGA-KIRP experiments. This approach differs from the subsection 4.5, where a single model is trained based on Mean Absolute Error (MAE) and all metrics are derived from this one model. For the TCGA-KIRP dataset, our data processing aligns with the approach outlined in Vale-Silva & Rohr (2021) The TCGA-KIRP experiments primarily use dense layers with ReLU activation for all modalities, except for the image data. The image modality is transformed into a feature vector using a pre-trained InceptionV3 (Szegedy et al., 2016) model with ImageNet weights (Deng et al., 2009). Unlike the CMU MOSI experiments, a single model is trained for survival prediction, from which all metrics are derived.

Both datasets employ simple dense-layer models built on these encoded representations for each modality. The specific point at which fusion occurs depends on the baseline method in consideration.

**Implementation:** The experimental settings differ slightly between the CMU MOSI and TCGA-KIRP datasets. For CMU MOSI, models are trained for 50 epochs with a batch size of 512, whereas, for TCGA-KIRP, the models run for 5 epochs with a batch size of 256. All experiments are conducted on an Nvidia©T4 GPU to ensure uniformity in computational resources. In both cases, multiple initializations are executed using ten different random seeds, permitting an assessment of not just average performance but also the volatility of predictions, a crucial metric, given the small-scale nature of the models involved. The standard deviation alongside the average of performance metrics is reported to offer a nuanced view of each method's robustness in tables 1 and 2.

Table 1: Results on the CMU-MOSI dataset (10 random seeds)

| Model | 2-classes | 7-classes | Regression |
|---|---|---|---|
| Early fusion | $0.627 \pm 0.016$ | $0.217 \pm 0.042$ | $1.298 \pm 0.083$ |
| Late fusion | $0.659 \pm 0.017$ | $0.282 \pm 0.017$ | $1.181 \pm 0.030$ |
| Max fusion | $0.661 \pm 0.015$ | $0.273 \pm 0.014$ | $1.270 \pm 0.034$ |
| Mean fusion | $0.659 \pm 0.018$ | $0.284 \pm 0.011$ | $1.214 \pm 0.024$ |
| Sum fusion | $0.645 \pm 0.017$ | $0.278 \pm 0.019$ | $1.304 \pm 0.114$ |
| Sequential fusion | $0.653 \pm 0.027$ | $0.274 \pm 0.009$ | $1.190 \pm 0.024$ |
| OSF | $\mathbf{0.687 \pm 0.013}$ | $\mathbf{0.300 \pm 0.015}$ | $\mathbf{1.139 \pm 0.023}$ |

A grid search in the set $\{1e^{-k}\}$, $k \in \{0, ..., 7\}$ is conducted to identify the optimal settings for OSF's custom loss function. We fix all L1 terms to the same value and all L2 terms to the same value to reduce the computation cost of the grid search. This tuning is unique to OSF and aimed at maximizing its performance.

The models are initially evaluated against basic fusion methods to show the advantages of OSF. Subsequently, OSF is integrated into a more complex model—Multimodal Infomax(Han et al., 2021)—and its performance is examined as a part of a fully developed multimodal system. An ablation study comparing OSF to a simplified version without the custom loss function is conducted, but will be detailed in a separate subsection to shed light on the individual contributions of OSF's components.

## 4.4 RESULTS AND IMPLICATIONS

**Results:** Our experiments provide strong evidence for the efficacy of OSF across different tasks. Importantly, OSF consistently outperforms traditional fusion techniques, offering substantial improvements over Sequential Fusion and other existing methods.

In the context of the CMU-MOSI dataset, OSF establishes itself as the best-performing model across all three evaluation tasks as detailed in Table 1. For the 2-class classification, OSF secures an accuracy of $0.687 \pm 0.013$, which surpasses the performance of other standard methods. Similarly, for the 7-class classification and regression tasks, OSF continues to dominate. The lower standard deviations in the performance metrics across all tasks suggest that OSF not only delivers better performance but also ensures more stable and reliable models.
Our findings on the TCGA-KIRP dataset (Table 2) are especially compelling. Unlike other methods that hover around the randomness threshold in terms of C-index, both OSF and Sequential Fusion significantly exceed it. Specifically, the C-index for OSF is $0.691 \pm 0.100$, clearly demonstrating its superiority in making meaningful predictions. Sequential Fusion also performs honorably, substantiating the utility of our proposed fusion methods. In terms of IBS, both our methods also overperform the baselines.

**Implications and General Observations:** The results are in perfect harmony with our initial hypotheses: an increase in the number of modalities and the inclusion of a custom loss function significantly enhance the performance of our models. This is particularly salient in the context of healthcare, where data is inherently multimodal, and traditional methods have often resorted to simplistic fusion mechanisms.

## 4.5 MODEL INTEGRATION

To demonstrate the adaptability and effectiveness of our proposed methods—Sequential Fusion and OSF—we integrated them into the Multimodal Mutual Information Maximization (MMIM) framework (Han et al., 2021), a well-established model noted for its strong performance on the CMU-MOSI dataset. Our objective was twofold: to ascertain the ease of integrating our methods into a fully developed and optimized model, and to evaluate their performance impact.

In replicating the MMIM framework, we adhered to the same data processing methods, experimental setup, and other implementation details as in the original paper. Importantly, the Sequential Fusion

Table 2: Results on the TCGA-KIRP dataset (10 random seeds)

| Model | C-index | IBS |
|---|---|---|
| Early fusion | $0.529 \pm 0.006$ | $0.484 \pm 0.159$ |
| Late fusion | $0.528 \pm 0.005$ | $0.475 \pm 0.147$ |
| Max fusion | $0.531 \pm 0.001$ | $0.500 \pm 0.132$ |
| Mean fusion | $0.531 \pm 0.001$ | $0.374 \pm 0.159$ |
| Sum fusion | $0.530 \pm 0.001$ | $0.406 \pm 0.135$ |
| Sequential fusion | $0.662 \pm 0.123$ | $0.264 \pm 0.095$ |
| OSF | $\mathbf{0.691 \pm 0.100}$ | $\mathbf{0.213 \pm 0.005}$ |

Table 3: Results on the CMU-MOSI using MMIM framework (20 random seeds)

| Model | 2-classes | 7-classes | Regression |
|---|---|---|---|
| MMIM | $0.826 \pm 0.006$ | $0.458 \pm 0.012$ | $0.723 \pm 0.013$ |
| MMIM + Sequential fusion | $0.829 \pm 0.010$ | $0.456 \pm 0.014$ | $0.725 \pm 0.014$ |
| MMIM + OSF | $\mathbf{0.831 \pm 0.005}$ | $\mathbf{0.462 \pm 0.016}$ | $\mathbf{0.719 \pm 0.015}$ |

mechanism was integrated into MMIM without any major changes to its architecture. For OSF, the contrastive part of the conventional MMIM was removed not to interfere with our orthogonal loss. This led to two notable impacts: first, an improvement in performance; and second, a reduction in training time per epoch from 75s to 45s on an NVIDIA©T4 GPU.

Table 3 presents the results of the MMIM framework and its variations, including Sequential Fusion and OSF. OSF exhibited superior performance across all evaluation metrics, both in terms of mean and best values. In contrast, Sequential Fusion showed comparable performance to the baseline in terms of mean. However, it excelled in producing best trajectories, as indicated by Table 4.

These findings underscore not only the ease with which our proposed methods can be integrated into existing frameworks but also their potential to improve overall model performance—especially in computational efficiency, as evidenced by the reduced training time for OSF but also in terms of evaluation metrics. This flexibility and effectiveness makes OSF a promising option for enhancing multimodal fusion in a wide range of applications.

## 4.6 ABLATION STUDY

Our ablation study serves as a comprehensive examination to contrast the performance of OSF and Sequential Fusion. Across all experiments, both OSF and Sequential Fusion perform admirably, but OSF consistently holds a slight edge. Examining the results from the CMU-MOSI dataset first, OSF outperforms Sequential Fusion across multiple metrics for both classification and regression tasks. However, it's important to note that Sequential Fusion also delivers robust results, showcasing its utility as a fusion technique, albeit below OSF. The narrative takes an interesting turn when evaluating performance on the TCGA-KIRP dataset. While OSF continues to excel, Sequential Fusion also achieves a high C-index value, which is particularly notable given that the baseline models almost hover around random performance. In this setting, Sequential Fusion stands as a significant improvement over basic fusion strategies, even if it doesn't quite reach the performance levels of OSF.

Table 4: Results on the CMU-MOSI using MMIM framework (best results)

| Model | 2-classes | 7-classes | Regression |
|---|---|---|---|
| MMIM | 0.838 | 0.477 | 0.702 |
| MMIM + Sequential fusion | **0.843** | 0.477 | 0.688 |
| MMIM + OSF | **0.843** | **0.488** | **0.682** |

When both fusion techniques are integrated into the MMIM framework, the performance trends remain consistent. OSF emerges superior, but Sequential Fusion is far from lagging behind. The fact that OSF shows improvements even when integrated into MMIM highlights its flexibility and adaptability. The superiority of OSF is very likely coming from its orthogonal loss, which ensures that the fused representations capture non-redundant and complementary features more effectively.

## 4.7 DISCUSSION

Our experiments demonstrate that OSF outperforms traditional fusion techniques, such as early fusion, late fusion, and mean/max/sum fusion, in various multimodal applications. By design, the structured approach to data integration of Sequential Fusion efficiently manages the volume and complexity of the data, preventing the model from being overwhelmed and effectively mitigating the curse of dimensionality. Adding the custom loss function encourages that each modality contributes unique and complementary information to the fused representation. Together, these methods offer valuable approaches to multimodal data fusion. While both are highly competent, OSF's orthogonal loss provides a distinct advantage in contexts with increased complexity and high dimensionality, complementing Sequential Fusion's sequential data integration strategy. As the number of modalities grows, the benefits of both sequential fusion and OSF become more pronounced. Given the strength of the OSF method in dealing with highly multimodal data, its application in fields like healthcare could potentially drive improvements in both existing and future systems. While OSF demonstrates promising results, there are potential limitations and areas for future research:

**Scalability to a large number of modalities:** Our current approach has been tested on applications involving a limited number of modalities. The performance and efficiency of OSF when applied to problems with a large number of modalities warrant further investigation.

**Hyperparameter optimization:** OSF introduces additional hyperparameters, such as the fusion order and orthogonal weights. Future work could explore techniques for optimizing these hyperparameters search to further improve the fusion process and model performance.

**Exploring all aspects of OSF:** Our experiments are limited to pairwise fusion and do not explore OSF's capability to incorporate expert knowledge when ordering the modalities.

**Exploring other orthogonalization techniques:** Our current method relies on a custom loss to incentivize the orthogonality of latent representations. Alternative orthogonalization techniques could be explored to potentially improve the fusion process of the resulting models.

## 5 CONCLUSION

In this paper, we introduce Orthogonal Sequential Fusion (OSF), a new fusion paradigm for multimodal machine learning. Our approach relies on two major contributions: a sequential mechanism for processing modalities and an orthogonal loss function. The sequential mechanism provides an unprecedented level of control over modality combination, enabling a precise representation of intrinsic intermodal relationships. The orthogonal loss function, through its exploitation of data complementarities and redundancies, amplifies the extraction of valuable insights from the multimodal context. Collectively, these innovative components establish OSF as a notable alternative to traditional fusion techniques in multimodal machine learning.

Throughout comprehensive experimental testing, OSF consistently outperforms traditional fusion strategies such as early fusion, late fusion, and mean/max/sum fusion. It differentiates itself not only with superior performance, but also with a more adaptable approach. This blend of flexibility and performance solidifies OSF as a viable and promising alternative to existing fusion paradigms. The empirical evidence underscores OSF's potential to significantly enhance the performance of multimodal machine learning models, demonstrating a unique combination of adaptability, explainability, and improved model performance.

While our approach's promising results underscore its effectiveness, we acknowledge certain limitations and potential avenues for future work. Key challenges include the scalability of our method to a larger number of modalities, its manageability given the high number of hyperparameters, and the prospect of exploring alternative orthogonalization techniques. Addressing these challenges could further improve the performance, interpretability, and adaptability of OSF, thereby laying the groundwork for effective and explainable multimodal machine learning models.

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

# A  CONVENTIONAL FUSION METHODS

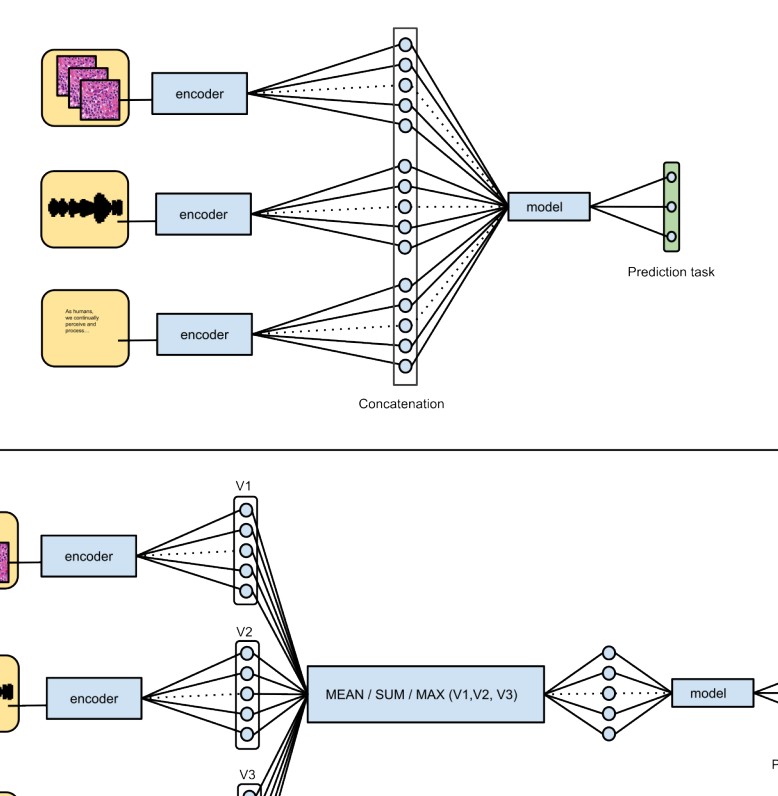

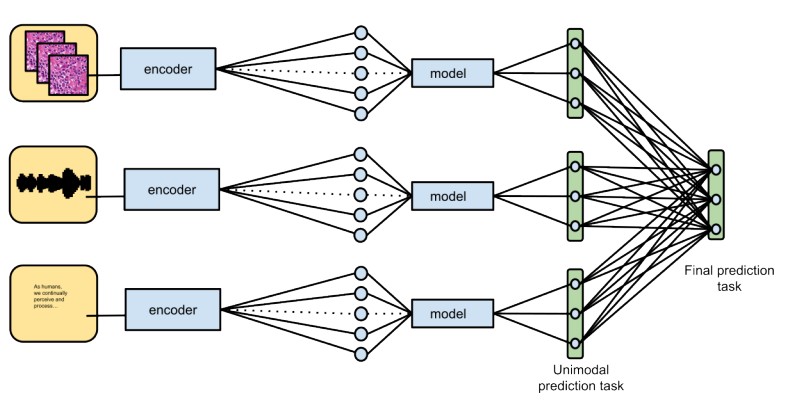

Figure 2: Early fusion, Mean/Max/Sum fusion, Late fusion with 3 modalities

