# OpenReview forum: "Orthogonal Sequential Fusion in Multimodal Learning"
_ICLR.cc/2024/Conference — ICLR 2024 Conference Withdrawn Submission_

### Official Review · Reviewer_8gNn · 2023-10-23

**Soundness:** 3 good
**Presentation:** 3 good
**Contribution:** 2 fair
**Rating:** 3
**Confidence:** 4

**Summary:**

This paper proposes a multimodal fusion paradigm called Orthogonal Sequential Fusion (OSF) to intergate in a sequential and selective manner.   The authors claim that OSF can integrate more complementary information from the multimodal inputs leading to more balanced representations. Experimental results on commonly used benchmark support their claims.

**Strengths:**

- Multimodal fusion is an interesting research topic. It is important to ensure integrating complementary information from multiple sources.
- This paper is well written, the proposed OSF algorithm is well motivated and clearly explained.
- Model-agnostic fusion techniques can be easily deployed in various multimodal learning methods, and thus I think the proposed methods can be seamlessly deployed in many multimodal applications.

**Weaknesses:**

- It seems that the proposed OSF determine the order of fusing the modalities by ranking the unimodal performances. This ranking-based approach may result in additionally computational cost in my view. On the other hand, sometimes it is impossible to use the same model for unimodal training and further evaluate the unimodal performance. Thus using unimodal performance as a proxy of distinct contributions to the task at hand may not be well motivated.
- The proposed method is intuitively motivated. Although the authors present the underlying motivations in detail. More theoretical analysis is appreciated.
- The empirical results are not convinced enough for me. For example, in Table. 2, the proposed method OSF are only compared to some simple fusion methods. It will be more convincing for me if more strong baseline such as self-attention or recent [1] can be involved in the future.

[1] Ma, Huan, et al. "Trustworthy multimodal regression with mixture of normal-inverse gamma distributions." NIPS 2021

**Questions:**

Please address the issues raised in the weakness section.

---

### Official Review · Reviewer_SAr6 · 2023-10-30

**Soundness:** 1 poor
**Presentation:** 1 poor
**Contribution:** 1 poor
**Rating:** 1
**Confidence:** 5

**Summary:**

This paper deals with mutimodal fusion in the context of neural networks. The proposed method is based on a sequential fusion of modalities. Modalities are processed iteratively in a predefined order. At each fusion layer, a specific loss metric is proposed that looks for having orthogonality between the two input embeddings.

Experimental results are reported on 2 datasets CMU-MOSI and TCGA-KIRP and compared to baseline simple fusion techniques.

**Strengths:**

No real novelty in the proposed work

**Weaknesses:**

This paper lack of reference to more advanced fusion technique. For example, what about [1, 2] works that present some alternate direction of analysis to deal with relative importance of modalities.

Some elements of the paper would need better clarification to have a better understanding. For example: how do embeddings of modalities are aligned to the same dimension D? What is the 'embedding' operation mentionned in Algorithm 1 performed after the CONCAT operation ?

In section 4.3 it is mentioned that input models are pre-trained models. So if they are not modified what is the effective impact of orthogonality criterion? For first level, if input modalities does not change then the orthogonal loss term will not change. For the other layers, we could expect 'embedding operation' (e.g. considering that it is a linear layer) of previous layer to guarantee having orthogonality with the second input layer while keeping same level of information.

When comparing to [2], what is the rationale behind using orthognality criterion rather than a more complex criterion such used in [2]?

Looking at Table 1, sequential approach turns out to be less performant than Late fusion. Could you elaborate on that? If the embedding operation consist of a linear layer, then we could consider that sequential fusion is quite similar to late fusion (using a linear fusion operator). Why is it so different in table 2?
Furthermore when considering confidence intervals in tables 2 and 3, there is no significant difference between results.

Ablation study provided is rather a discussion on previous results than additional analysis with additional experiments.

[1] Li, F., Neverova, N., Wolf, C., & Taylor, G. (2016). Modout: Learning to Fuse Modalities via Stochastic Regularization. Journal of Computational Vision and Imaging Systems, 2(1).

[2] Andrew, G., Arora, R., Bilmes, J., Livescu, K.: Deep canonical correlation analysis.In: International Conference on Machine Learning. (2013) 1247–1255

**Questions:**

1. What is the 'embedding' operation in Algorithm 1? (performed just after CONCAT to get a D-dimensional fused embedding)
2. What about orthogonality criterion rather than cross correlation criterion such as in [2]?
3. Since CMU-MOSI has only 3 modalities, what is the real impact of modality ranking?
4. Why does the sequential fusion approach performs worse than late fusion?

---

### Official Review · Reviewer_HooP · 2023-10-31

**Soundness:** 2 fair
**Presentation:** 2 fair
**Contribution:** 2 fair
**Rating:** 3
**Confidence:** 3

**Summary:**

This paper proposes a new fusion paradigm called orthogonal sequential fusion. It sequentially merges inputs and permits selective weighting of modalities. It offers a flexible way to weight and prioritize individual modalities based on their relevance, allowing for dynamic weighting or prioritization of individual modalities. Moreover, it explores the potential of orthogonal representations in fusion and is used to improve fusion performance.

**Strengths:**

1. It proposes a new sequential fusion paradigm, which is significant different from existing traditional fusion paradigms.
2. It designs a method to determine the optimal order for sequential fusion.
3. The article is easy to understand and the results show superiority to the traditional early fusion, late fusion, max/mean/sum fusion and sequential fusion.

**Weaknesses:**

1. The related work lacks a discussion on multimodal information fusion works in recent years. It lacks the analysis of the state-of-art methods.
2. This paper proposes a way for modalities ordering. It starts from the least performing modality and moves towards the most performing one. However, the soundness of this approach does not have good theoretical support. The reason that it can provide valuable insights into the relationships and interactions between different modalities is unclear.
3. Orthogonal orthogonality is used in the fusion of two modalities and is not closely related to the proposed sequential fusion paradigm.
4. It lacks the experiment to validate the superiority of the modalities ordering way to other ordering results.

**Questions:**

1. When fusing N modalities, the proposed method needs N-1 fusion layers. Compared to Early fusion, Mean/Max/Sum fusion, Late fusion, will such a N-1 fusion layer increase computational complexity and runtime?
2. The loss function promotes the orthogonal representations of features. But how the information of the two modalities is fused after determining the orthogonal representations?

---

### Official Review · Reviewer_ZKms · 2023-11-01

**Soundness:** 2 fair
**Presentation:** 2 fair
**Contribution:** 2 fair
**Rating:** 5
**Confidence:** 2

**Summary:**

The authors introduced OSF, a novel fusion paradigm for multimodal machine learning. This fusion approach allows features from different modalities to be fused step-by-step. The orthogonality facilitation model extracts as much complementary information as possible from the different modalities. The method performs well compared to existing fusion methods and can be incorporated into a state-of-the-art model.

**Strengths:**

1. The fusion paradigm proposed by the authors in this paper is straightforward in its thinking. Compared to the traditional Multimodal Learning and Traditional Fusion Techniques, the OSF proposed in this paper performs an initial ordering of modalities through the relative strengths of different modalities in the task and fuses them step by step.
2. The author added an orthogonal term to the loss function, which allows the model to extract complementary features from different modes to the maximum extent possible.
3. The OSF method proposed in this paper outperforms a variety of previous traditional Multimodal Learning and Traditional Fusion Techniques in terms of experimental metrics. and is able to be added to existing multimodal fusion models.

**Weaknesses:**

1. The explanation of method theory in Chapter 3 is slightly weak. This chapter is the core work of the article, and the author should provide a more detailed explanation of this part of the work.
2. In Section 3.2, the authors described the OSF method “fuse the modalities starting from the least performing modality and moving towards the most performing one.” I would like to know why the authors chose to do the ordering in this way, and how the fusion results would change if the order was reversed?
3. In the same section, the author mentions that " In some cases, certain modalities may be highly correlated or redundant, and the ranking may not accurately reflect their true contributions to the task. In such cases, alternative methods for determining fusion order, such as using expert knowledge, may be more suitable." How should the user judge whether OSF can be used for this task, and is there a clear evaluation metric to assist the user in this judgment?

**Questions:**

Please refer to weaknesses